# Ultrasonic-Assisted Synthesis of Benzofuran Appended Oxadiazole Molecules as Tyrosinase Inhibitors: Mechanistic Approach through Enzyme Inhibition, Molecular Docking, Chemoinformatics, ADMET and Drug-Likeness Studies

**DOI:** 10.3390/ijms231810979

**Published:** 2022-09-19

**Authors:** Ali Irfan, Ameer Fawad Zahoor, Shagufta Kamal, Mubashir Hassan, Andrzej Kloczkowski

**Affiliations:** 1Department of Chemistry, Government College University Faisalabad, Faisalabad 38000, Pakistan; 2Department of Biochemistry, Government College University Faisalabad, Faisalabad 38000, Pakistan; 3The Steve and Cindy Rasmussen Institute for Genomic Medicine, Nationwide Children’s Hospital, Columbus, OH 43205, USA; 4Department of Pediatrics, The Ohio State University, Columbus, OH 43205, USA

**Keywords:** ultrasonic-assisted green synthesis, furan-oxadiazole molecules, tyrosinase inhibitors, chemoinformatics, molecular docking, ADMET study, structure-activity relationship

## Abstract

Furan-oxadiazole structural hybrids belong to the most promising and biologically active classes of oxygen and nitrogen containing five member heterocycles which have expanded therapeutic scope and potential in the fields of pharmacology, medicinal chemistry and pharmaceutics. A novel series **5a–j** of benzofuran-oxadiazole molecules incorporating S-alkylated amide linkage have been synthesized using ultrasonic irradiation and screened for bacterial tyrosinase inhibition activity. Most of the synthesized furan-oxadiazole structural motifs exhibited significant tyrosinase inhibition activity in the micromolar range, with one of the derivatives being more potent than the standard drug ascorbic acid. Among the tested compounds, the scaffold **5a** displayed more tyrosinase inhibition efficacy IC_50_ (11 ± 0.25 μM) than the ascorbic acid IC_50_ (11.5 ± 0.1 μM). Compounds **5b**, **5c** and **5d** efficiently inhibited bacterial tyrosinase with IC_50_ values in the range of 12.4 ± 0.0–15.5 ± 0.0 μM. The 2-fluorophenylacetamide containing furan-oxadiazole compound **5a** may be considered as a potential lead for tyrosinase inhibition with lesser side effects as a skin whitening and malignant melanoma anticancer agent.

## 1. Introduction

The discovery of tyrosinase is linked with color changes, such as browning of fruits and change of color from blue to red and then further to brown or black in mushrooms *Russulafoetens* and *Rhizopusnigricans* due to oxidation processes, as revealed in 1885 by Bourquelotet. Bertrand named the new isolated enzyme from *Rhizopusnigricans* as tyrosinase which is responsible for the oxidation of tyrosine and proven to be a precursor of melanin biosynthesis. Polyphenol oxidase (PPO) and monophenoloxygenase are multi-copper-based glycoprotein membranes bound metalloenzymes, called tyrosinases, which are present in a wide variety of organisms such as fungi, bacteria, plants and humans [1,2,3,4,5]. The epidermal basal layer contains melanocytes in which tyrosinase catalyzes melanin formation in a multistep synthetic process. The type of skin color of any organism depends on the melanin type, its pattern of distribution and the amount of melanin in the keratinocytes. The binuclear copper tyrosinase catalyzes the two prominent and distinct biosynthetic reactions which involves orthrohydroxylation of monophenols to orthrodiphenols termed as monophenolase activity; and the conversion of orthrodiphenols to orthro-quinones via oxidation process called diphenolase activity, as depicted in Figure 1 [4,5,6]. Tyrosinase also catalyzes the formation of neuromelanins, while the excessive conversion of dopamine from the oxidation of dopaquinones results in neuronal disorders which link tyrosinase with neurodegenerative diseases such as Parkinson’s disease [7,8,9].

Melanin formation in skin is essential for protective action against exposure to UV radiation, which is a major cause of skin cancer, while melanin formation in excessive amounts can affect and cause diseases such as solar lentigo, cancer, ephelis, melanoderma, age spots, melasma, inflammatory flecks and malignant melanoma. Pigmented acne scars and freckles are major concerns of cosmetology [10,11]. In addition, agricultural products, such as bruised vegetables and fruits, are prone and vulnerable to browning, especially at the post-harvest stage, leading to quality and freshness loss, faster degradation and shorter shelf-life. Generally, the tyrosinase enzymes extracted from different sources, such as animals, plants, fungi and bacteria, exhibit a broad spectrum of applications in various fields. Bacterial tyrosinase exhibits versatile applications in the fields of the dye industry, pharmaceuticals, agriculture and electronics. The PDB ID: 3NM8 of *Bacillus megaterium* tyrosinase can be used in directed evolution experiments, while *Bacillus thuringiensis* tyrosinase can be used as a water decontamination agent and as a pesticide [12,13]. Bacterial tyrosinase-based biosensors can be applied in the determination of phenolic compounds in samples. The quantity of toxic cyanogenic glycosides can be determined in various foods, such as cherries, apricot kernels, peaches, etc., by the application of disposable biosensors [14,15]. Bacterial tyrosinase has wide a spectrum of applications such as in the production of L-Dopa for Parkinson’s disease, melanin production, biocatalysis, bioengineering, phenol and dye removal, protein cross-linking, etc. [16,17]. Despite the useful applications of tyrosinase, it can cause melanoma cancer, inflammatory flecks and other skin diseases. Therefore, tyrosinase inhibitors are widely utilized by dermatologists and in agriculture, food and cosmetic industries. The development of novel tyrosinase inhibitors can produce improved alternatives to cure hyper-pigmentation disorders, protect skin and preserve food quality [18,19]. From this perspective, heterocyclic compounds, especially those containing azole frame along with carbonyl moiety, are an attractive class of high potency tyrosinase inhibitors [20,21,22]. The most well-known group of the azole family is oxadiazole which is part of several clinical drugs, as shown in Figure 2. A series of studies revealed that structural motifs with oxadiazole cores display a wide array of biological and pharmacological activities, such as alkaline phosphatase inhibitors and antibacterial, antifungal and antiviral agents, as well as anti-inflammatory, antihypertensive, anticancer, antidiabetic and tyrosinase inhibitors [23,24,25,26].

Recent studies of furan core-based natural and synthetic derivatives (Figure 3) have drawn considerable interest from synthetic and medicinal scientists due to the profound physiological potential and broad spectrum of chemotherapeutic efficacy against a wide variety of diseases and pathogens [27,28,29]. Furan derivatives have been evaluated for diverse and versatile therapeutic applications such as antibacterial, antiviral and antifungal agents, laccase catalysts, antitumor, antiproliferative, hemolytic, thrombolytic and inflammation inhibitors, analgesic and antihyperglycemic drugs and antioxidants [30,31,32,33,34,35,36,37,38]. The keto- and carboxy-furan derivatives and furan-carboxamide-containing oxoquinazolin scaffolds proved to be promising tyrosinase inhibitors [39,40,41]. In the present research, our group reported the ultrasonic-assisted green synthetic methodology of furan-bearing oxadiazole scaffolds, their spectroscopic characterization, biological screening as tyrosinase inhibitors, molecular docking and absorption, distribution, metabolism, excretion, and toxicity (ADMET) studies and cheminformatics analyses of novel benzofuran-oxadiazole structural motifs. Faiz et al. reported the conventional synthesis of these benzofuran-oxadiazole structural hybrids which required a much longer time for completion and achieved low yields as compared to the ultrasonic-assisted methodology [30,31,32].

## 2. Results and Discussion

### 2.1. Synthesis of Furan-Oxadiazole Derivatives **5a–j**

The ultrasonic-assisted synthetic strategy for the synthesis of the target oxadiazole moiety containing furan molecules has been depicted in Figure 1 [30,31,32]. In this multistep methodology, ethyl benzofuran-2-carboxylate was obtained following the synthetic procedure reported by Kowalewska et al. involving the reaction of bromo-substituted salicylaldehyde **1** and ethyl chloroacetate in the presence of KOH which acted as a basic dehydrating agent [42]. In the next step, benzofuran-2-carbohydrazide was obtained by refluxing benzofuran-2-carboxylate with hydrazine hydrate in methanol which, upon further treatment with CS_2_/KOH, resulted in thiol-functionality-containing oxadiazole-based furan molecules **2 [43,44]**. On the other side, the aliphatic and aromatic amines were reacted with bromoacetyl bromide to obtain substituted bromoacetanilide derivatives **4a–j [40]**. The substituted S-alkylated oxadiazole-based furan structural hybrids **5a****–j** with chemical structures shown in Table 1 were obtained with moderate to good yields (53–79%) with treatment of substituted bromoacetanilide derivatives **4a–j** with oxadiazole-based furan derivatives under basic ultrasonic-assisted conditions. All the synthesized structural motifs **5a–j** were examined by utilizing various spectroscopic approaches such as ^1^H NMR, ^13^C NMR and high-resolution mass spectrometry (HRMS). The derivative bromobenzofuran-oxadiazole based 2-chlorophenyl acetamide **5b** was obtained with minimal 53% yield, while the bromobenzofuran 2-methoxyphenyl acetamide 5f was obtained with a maximal 79% yield. The NMR spectrum data of all synthetic compounds have been mentioned in Appendix A.

### 2.2. Bacterial Tyrosinase Inhibition and SAR

The target derivatives were evaluated against tyrosinase enzyme (indigenously isolated from Bacillus subtilis NCTC 10400) inhibition [45,46]. The synthesized derivatives showed activities with IC_50_ values ranging from 11 ± 0.25 µM to 49.5 ± 0.9 2 µM which are almost comparable to the well-known tyrosinase inhibitor—ascorbic acid [47,48,49]—as shown in Table 2. The substituents and the position of the substituents on the phenyl ring attached with the S-alkylated amide linkage significantly increase or decrease the inhibitory potential of synthesized scaffolds **5a****–****j** depending on the nature of substituent. The results from Table 2 revealed that electron-withdrawing (EWD) halogen substituents on the phenyl group reveal promising inhibitory effects. The highest electronegative fluoro group at the ortho position on the phenyl ring displayed the best tyrosinase inhibition efficacy, higher than the reference standard inhibitory drug ascorbic acid, while the third, fourth and fifth positions were less active than the second position on the phenyl ring, as depicted in Figure 4. However, among all the synthesized derivatives, the scaffold **5a** exhibited the highest tyrosinase inhibition potential with IC_50_ value of 11 ± 0.25 µM, this effect results from the presence of the electron-withdrawing (EWD) fluoro group. We observed that halo-substituted derivatives **5a****–****5d** exhibited better inhibitory effects than **5e–5j** motifs containing electron-denoting (EDG) alkyl (ethyl, methyl and methoxy) substituents on the phenyl ring. The analogues **5b** with 2-chloro, **5c** with 3,4-dichloro and **5d** with 4-fluoro substituents showed remarkably good IC_50_ values of 12.4 ± 0.0, 12.7 ± 0.0 µM and 15.5 ± 0.0, respectively, due to the presence of EWD functionalities as displayed in Figure 4. Moderate tyrosinase inhibitory potential was displayed by **5e** with 3,4-dimethyl, **5f** with 2-methoxy**, 5g** with phenyl and **5h** with 2,4-dimethyl substituents exhibited by IC_50_ values of 25 ± 0.75, 27 ± 1.00, 30 ± 1.50 and 36 ± 0.25, respectively, as shown in Appendix A. It was noted that among the synthesized derivatives, the least effective structural hybrids were **5i** with 2,5-dimethoxy (EDG) substituents and **5j** with a diethyl group leading to decreases in inhibition potential with IC_50_ values of 48 ± 0.96 and 49.5 ± 0.92, respectively, as depicted in Appendix A. The graphical representation clearly indicates that the higher the IC_50_ value, the lesser the tyrosinase inhibition efficacy and vice versa. Structure-activity relationship (SAR) studies indicate that overall substituent-related tyrosinase activity decreases in the following way: 2-flouro > 2-chloro > 3,4-dichloro > 4-fluoro > 3,4-dimethyl > 2-methoxy > phenyl > 2,4-dimethyl > 2,5-dimethoxy > diethyl.

### 2.3. Structural Assessment of Bacterial Tyrosinase

Bacterial tyrosinase from Bacillus megaterium (PDB ID 3NM8) is a copper-containing enzyme consisting of two chains having 303 residues [50,51]. The detailed overall structural assessment of tyrosinase shows that it is comprised of α-helices, β-sheets and coil in 34%, 12% and 53%, respectively, while its crystallographic resolution is 2.00 Å, with an R-value of 0.273. X-ray crystallography provides the dimensions of the unit cell. It appears that 98% of protein conformations were found in preferential sections, while 100.0% lies in the permitted part of the Ramachandran graph (Figure 5).

### 2.4. RO5 Validation of Newly Designed Furan-Oxadiazole Ligands

The newly designed chemical compounds were evaluated by us for drug-likeness using Lipinski’s rule of five (RO5) and then cheminformatics methods, which are considered major hallmarks in drug development processes. The furan-oxadiazole compounds **5a****–****j** were tested by comparing their RO5 values with standard values of orally active drugs in humans to verify their lead-like behavior. According to Lipinski’s rule of five, an orally active drug should satisfy the following criteria: its molecular mass, an octanol-water partition coefficient (logP), the number of hydrogen bond acceptors (HBA) and the number of hydrogen bond donors (HBD) must have values less than 500 (g/mol), 5, 10 and 5, respectively. The molecular weights (g/mol) of all the synthesized furan-oxadiazole structural motifs **5a****–****j** satisfy Lipinski’s requirement (<500 g/mol) as shown in Table 3. The analysis of octanol-water partition coefficients logP for all furan-oxadiazole scaffolds **5a****–****5j** shows that logP values for **5c** (5.88) and **5e** (5.29) exceeded than standard value (< 5), while the remaining furan-oxadiazole derivatives have values that conform with the requirement (logP < 5), see Table 3. The novel furan-oxadiazole derivatives **5a****–****j** were also evaluated on the basis of two important Lipinski’s rule screening parameters HBA and HBD. The results shown in Table 3 indicate that all furan-oxadiazole derivatives contain less than 10 hydrogen bond acceptors and less than 5 for hydrogen bond donors, satisfying Lipinski’s requirements HBA < 10 and HBD < 5. The last column in Table 3 shows which compounds satisfy all four rules of five (RO5). Another important parameter in the drug development process is the total polar surface area (PSA) defined as the surface sum over all polar atoms or molecules, primarily oxygen and nitrogen, also including their attached hydrogen atoms. PSA is used to determine the optimum ability of a drug to permeate cells. To penetrate the blood–brain barrier a PSA less than 89 Å^2^ is usually needed [52,53,54]. The analysis of our newly synthesized furan-oxadiazole compounds **5a–j** shows that all of them have PSA values < 89 Å^2^_,_ which clearly indicates and supports their lead-like behavior as seen in Table 3.

### 2.5. Molecular Docking Studies of Furan-Oxadiazoles **5a–j**

Our computational approach included molecular docking simulations using AutoDock software to predict the conformations of newly designed furan-oxadiazole ligands in the binding sites of proteins and their binding energies [55,56]. The docking energy values ∆G_binding_ for all the docked furan-oxadiazole complexes were calculated from Equation (1).
∆G_binding_ = ∆G_gauss_ + ∆G_repulsion_ + ∆G_hbond_ + ∆G_hydrophobic_ + ∆G_tors_(1)

Here ∆G_gauss_ is the term representing two Gaussian functions; ∆G_repulsion_ is the square of the distance if closer than a threshold value; interactions of metal ions and hydrogen bonds are represented by a ramp function ∆G_hbond_; ∆G_hydrophobic_ is also a ramp function and ∆G_tors_ represents the torsional term proportional to the number of rotatable bonds. The predicted docking energy values for all designed furan-oxadiazole ligands **5a–j** (in kcal/mol) are listed in Table 4 and compared with the standard drug—ascorbic acid binding energy value −6.6 kcal/mol.

The typical error for AutoDock is described as −2.5 kcal/mol (http://autodock.scripps.edu/). The furan-oxadiazole synthesized ligands **5a****–****j** have a unique chemical scaffold with different substitutions at peculiar positions which results in less fluctuation in docking energy. Therefore, docking energy values among all the synthesized furan-oxadiazole ligands do not vary greatly from the −2.5 kcal/mol value. The docking energy values show that all synthesized scaffolds **5a****–****j** have good docking energy values in comparison with standard drug ascorbic acid.

#### Binding Pocket and Hydrogen (H) Binding of Furan-Oxadiazole **5a**

Bromobenzofuran-oxadiazole-based 2-fluorophenyl acetamide derivative **5a** was selected as the most potent and promising bacterial tyrosinase inhibitor based on in vitro results. Binding analysis showed that scaffold **5a** developed a stable docking complex by forming a three hydrogen and one hydrophobic bond with His-208, His-60 and Met-61, respectively. There is a weak interaction between the benzene ring of **5a** and His-208 with a bond distance 2.80 Å. Similarly, the Sulphur (S) atom forms a good binding interaction with appropriate binding distance (2.71 Å) with Met-61, whereas oxadiazole moiety developed another hydrogen linkage with Met-61 with a bond distance of 2.75 Å. The backbone (carbon chain) of **5a** forms another bond with His-60 with a good binding interaction distance of 2.80 Å. The overall docking results show that furan-oxadiazole derivative **5a** binds exclusively to the active section of the target area in the protein where the copper atoms are present, which may ensure the competitive behavior of the designed furan-oxadiazole ligand **5a**. In agreement with an already reported study which describes the importance of the presence of binding pocket residues in the downstream signaling pathways, our docking results for furan-oxadiazole derivative **5a** show good correlation with these data as depicted in Figure 6A,B [57].

### 2.6. ADMET and Drug-Likeness Studies of Furan-Oxadiazoles **5a–j**

Analysis of the physicochemical and pharmacokinetic properties of synthesized furan-oxadiazole compounds **5a****–****5j** using absorption, distribution, metabolism, excretion and toxicity (ADMET) studies revealed that furan-oxadiazoles have good gastrointestinal absorption, and all of the compounds in this investigation have bioavailability values of 0.55, which suggest that these scaffolds have promising futures as pharmaceuticals. Generally, a bioavailability score of a minimum of 0.10 is required for a compound to be considered a drug candidate. These compounds also have good aqueous estimated solubility values (ESOL) (Log S) and acceptable lipophilic properties (iLog P). The drug-likeness studies indicate that all synthesized scaffolds completely follow Lipinski’s rule of five for drugs and have acceptable topological surface areas. These compounds were non-substrates of P-glycoprotein (P-gp) which is a transporter protein of cell membranes and controls the efflux of substances from cells. Therapeutic drugs that are P-gp substrates might be pushed out of the cells by P-gp, which impedes its absorption, permeability and retention in cells. Furthermore, the toxicity studies of these compounds showed that these compounds are non-carcinogenic, non-AMES toxic and are non-inhibitors of the hERG potassium channel responsible for cardiac-action potential repolarization and do not interfere in its normal function. Moreover, the furan-oxadiazole compound **5a** shows good inhibitory activities and displays adequate and satisfactory pharmacokinetic properties. The compound **5a** exhibits good gastrointestinal (GI) absorption and bioavailability scores along with significantly good water solubility and lipophilic properties, and good topological polar surface area (TPSA). The drug-likeness studies of derivative **5a** confirm that this compound completely follows Lipinski’s drug rule of five and derivative **5a** is also non-AMES toxic, non-carcinogenic and non-substrate of P-gp protein. On the basis of satisfactory ADMET and drug-likeness properties, it can be concluded that furan-oxadiazole compound **5a** can be safely developed as a promising drug candidate against bacterial tyrosinase target enzymes. Generally, all furan-oxadiazole compounds **5a****–****5j** showed good ADMET and drug-likeness profiles which can be viewed in Table 5.

## 3. Materials and Methods

### 3.1. Synthesis and Characterization Techniques

The ultrasonic irradiated experimental synthetic strategy was performed in a 1.9-L capacity ultrasonic cleaner bath (model 1510) powered by a 115 V heater switch, 47 kHz and mechanical timer. In this research, all analytical grade materials, such as starting materials, reagents and solvents, were of Alfa Aesar, Merck, Fischer and Sigma Aldrich origin. All materials were purchased through local suppliers and analytical grade distilled solvents were used in the research. The reactions were monitored by thin layer chromatography (TLC) using aluminum-backed silica gel plates. Purification of the synthesized compounds was carried out using flash column chromatography, and compounds were further cleaned using recrystallization techniques. The 400 MHz proton NMR (δ = ppm) and 100 MHz carbon-13 NMR spectra (δ = ppm) were estimated in dimethyl sulfoxide-d6 (DMSO-d6), respectively, and spectrophotometer (Bruker model AV-400) was used to document both proton and carbon-13 spectra. The coupling constant (J) values are presented in hertz (Hz) while proton NMR and carbon-13 NMR spectra are characterized in the form of abbreviations such as s, d, t and m for singlet, doublet triplet and multiple, respectively.

### 3.2. Synthesis of Furan-Oxadiazole Scaffolds **5a–j** by Ultrasonic Irradiated Synthetic Approach

The scaffold furan-oxadiazole-2-thiol **2** was achieved by consecutives reactions as already reported [30,31,32]. In this methodology, 0.137 millimolar furan-oxadiazole-2-thiol **2** was dissolved in acetonitrile (15 mL), then 0.213 millimolar pyridine was added to the compound **2** solution as a basic catalyst and the reactants were stirred at 0 °C for 15 min (min). Substituted bromoacetanilides **4a–j** (0.24 millimolar) were slowly poured in the reaction vessel with continuous stirring and sonicated in a sonication bath at 40 °C for 30 min as depicted in Figure 1. The TLC technique was used to monitor reaction progress and after the completion of reaction, final substituted furan-oxadiazole products **5a–j** were obtained in the form of precipitates by adding petroleum ether. Washing of precipitated derivatives was carried out with distilled filtered water. The furan-oxadiazole products **5a–j** were purified by recrystallization in alcoholic solvents and column chromatography techniques.

#### 3.2.1. 2-((5-(5-Bromobenzofuran-2-yl)-1,3,4-oxadiazol-2-yl)thio)-N-(2-fluorophenyl)acetamide (**5a**)

Yield (55%), White Powder, MP (187–188 °C), ^1^H NMR (DMSO-*d_6_*; 400 MHz: 4.45 (s, 2H, CH_2_), 7.25−7.30 (m, 3H, Ar), 7.58−7.77 (m, 3H, Ar), 7.91 (d, *J* = 8.0 Hz, 1H, Ar), 8.03 (d, *J* = 2.0 Hz, 1H, Ar), 10.26 (s, exch., 1H, NH),^13^C NMR (DMSO-*d_6_*: 100 MHz): 36.7, 109.8, 114.1, 115.6, 115.8, 116.6, 124.0, 124.6, 125.2, 125.9, 126.2, 129.3, 130.1, 141.0, 153.9, 158.0, 164.2, 166.5. HR-MS Calcd.447.992[M+] (100%). Anal.Calcd.for: C_18_H_11_BrFN_3_O_3_S; C, 48.23; H, 2.47; N, 9.37; Found C, 48.28; H, 2.46; N, 9.42.


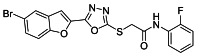
.

##### 3.2.2. 2-((5-(5-Bromobenzofuran-2-yl)-1,3,4-oxadiazol-2-yl)thio)-N-(2-chlorophenyl)acetamide (**5b**)

Yield (53%), White Powder, MP (197–199 °C), ^1^H NMR (DMSO-*d_6_*; 400 MHz): 4.46 (s, 2H, CH_2_), 7.22 (t, *J* = 8.0 Hz, 1H, Ar), 7.34 (t, *J* = 8.0 Hz, 1H, Ar), 7.51 (d, *J* = 8.0 Hz, 1H, Ar), 7.64−7.77 (m, 4H, Ar), 8.03 (d, *J* = 2.0 Hz, 1H, Ar), 10.12 (s, exch., 1H, NH). ^13^C NMR (DMSO-*d_6_*; 100 MHz): 36.6, 109.9, 110.2, 114.1, 116.6, 125.2, 126.0, 126.5, 127.8, 129.3, 129.7, 130.1, 134.5, 141.0, 153.9, 158.0, 164.1, 165.5.HR-MS Calcd. 463.954[M+] (100%). Anal.Calcd.for: C_18_H_11_BrClN_3_O_3_S; C, 46.52; H, 2.39; N, 9.09; Found C, 46.56; H, 2.33; N, 9.14.



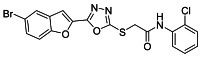



###### 3.2.3. 2-((5-(5-Bromobenzofuran-2-yl)-1,3,4-oxadiazol-2-yl)thio)-N-(3,4-dichlorophenyl)acetamide (**5c**)

Yield (77%), White Powder, MP (204–205 °C), ^1^H-NMR, DMSO-*d_6_*, 400 MHz (δ/ppm): 4.39 (s, 2H, CH_2_), 7.48 (d, *J* = 8.0 Hz, 1H, Ar), 7.58–7.64 (m, 2H, Ar), 7.66–7.73 (br s, 2H, Ar), 7.96 (s, 1H, Ar), 8.03 (s, 1H, Ar), 10.75 (s, exch., 1H, NH), ^13^C-NMR, DMSO-*d_6_*, 100 MHz (δ/ppm): 37.1, 109.9, 114.1, 116.6, 119.0, 120.5, 125.2, 125.3, 129.3, 130.0, 131.0, 131.1, 138.7, 141.0, 153.9, 158.0, 164.2, 166.4. HR-MS Calcd. 498.892 [M+] (100%). Anal. Calcd.for:C_18_H_10_BrC_l2_N_3_O_3_S; C, 43.31; H, 2.02; N, 8.42; Found C, 43.36; H, 2.06; N, 8.45.



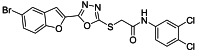



####### 3.2.4. 2-((5-(5-Bromobenzofuran-2-yl)-1,3,4-oxadiazol-2-yl)thio)-N-(4-fluorophenyl)acetamide (**5d**)

Yield (63%), White Powder, MP (192–193 °C), ^1^H NMR (DMSO-*d_6_*: 400 MHz): 4.38 (s, 2H, CH_2_), 7.16 (d, *J* = 8.5 Hz, 2H, Ar), 7.40 (dd, *J* = 8.0 and 2.0 Hz, 1H, Ar), 7.58−7.66 (m, 2H, Ar), 7.75 (d, *J* = 8.5 Hz, 2H, Ar), 8.03 (s, 1H,Ar), 10.44 (s, exch., 1H, NH), ^13^C NMR (DMSO-*d_6_*; 100 MHz): 37.1, 109.8, 114.1, 115.5, 115.7, 116.4, 121.1, 125.2, 127.0, 130.0, 138.9, 142.1, 154.6, 155.0, 164.4, 164.7.HR-MS Calcd. 448.954[M+] (100%). Anal Calcd.for: C_18_H_11_BrFN_3_O_3_S; C, 48.23; H, 2.47; N, 9.37; Found C, 48.26; H, 2.49; N, 9.43.



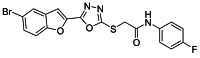



######## 3.2.5. 2-((5-(5-Bromobenzofuran-2-yl)-1,3,4-oxadiazol-2-yl)thio)-N-(3,4-dimethylphenyl)acetamide (**5e**)

Yield (65%), White Powder, MP (185–186 °C), ^1^H-NMR (DMSO-*d_6_*; 400 MHz): 2.16 (s, 3H, Me), 2.17 (s, 3H, Me), 4.35 (s, 2H, CH_2_), 7.05 (d, *J* = 2.0 Hz, 1H, Ar), 7.28−7.35 (m, 2H, Ar), 7.64−7.77 (m, 3H, Ar), 8.02 (d, *J* = 2.0 Hz, 1H, Ar), 10.18 (s, exch., 1H, NH),^13^C NMR (DMSO-*d_6_*; 100 MHz): 18.9, 19.7, 37.1, 109.8, 114.1, 116.6, 116.8, 120.5, 125.2, 129.3, 129.8, 130.0, 131.6, 136.4, 136.6, 141.4, 153.9, 158.0, 164.3, 164.4. HR-MS Calcd.458.095[M+] (100%).Anal.Calcd for: C_20_H_16_BrN_3_O_3_S; C, 52.41; H, 3.52; N, 9.17; Found C, 52.45; H, 3.53; N, 9.22.



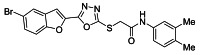



######### 3.2.6. 2-((5-(5-Bromobenzofuran-2-yl)-1,3,4-oxadiazol-2-yl)thio)-N-(2-methoxyphenyl)acetamide (**5f**)

Yield (79%), Off white powder, MP (188–190 °C), ^1^H NMR (DMSO-*d_6_*: 400 MHz): 3.84 (s, 3H, OMe), 4.45 (s, 2H, CH_2_), 6.91 (dd, *J* = 8.0 and 2.0 Hz, 1H, Ar), 7.05−7.12 (m, 2H, Ar), 7.64 (s, 1H, Ar), 7.66−7.77 (m, 2H, Ar), 7.95 (d, *J* = 8.0 Hz, 1H, Ar), 8.03 (d, *J* = 2.0 Hz, 1H, Ar), 9.72 (s, exch., 1H, NH),^13^C NMR (DMSO-*d_6_*; 100 MHz): 36.9, 55.8, 109.9, 111.4, 114.1, 116.6, 120.4, 121.7, 124.9, 125.2, 126.9, 129.3, 130.1, 141.1, 149.6, 153.9, 158.0, 164.2, 165.1. HR-MS Calcd.458.92 [M+] (100%). Anal.Calcd for: C_19_H_14_BrN_3_O_4_S; C, 49.58; H, 3.07; N, 9.13; Found C, 49.62; H, 3.12; N, 9.17.



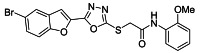



########## 3.2.7. 2-((5-(5-Bromobenzofuran-2-yl)-1,3,4-oxadiazol-2-yl)thio)-N-phenylacetamide (**5g**)

Yield (65%), White powder, MP (227–228 °C), ^1^H-NMR, DMSO-*d_6_*, 400 MHz (δ/ppm): 4.39 (s, 2H, CH_2_), 7.08 (t, *J* = 8.0, 1H, Ar), 7.33 (t, *J* = 8.0 Hz, 2H, Ar), 7.52–7.76 (m, 5H, Ar), 8.02 (d, *J* = 2.0 Hz, 1H), 10.45 (s, exch., 1H, NH),^13^C-NMR, DMSO-*d_6_*, 100 MHz (δ/ppm): 37.2, 109.8, 114.1, 116.6, 119.3, 123.8, 125.2, 129.0, 129.3, 130.0, 138.7, 141.0, 153.9, 158.0, 164.8.HR-MS Calcd. 428.972 [M+] (100%). Anal.Calcd for: C_18_H_12_BrN_3_O_3_S; C, 50.25; H, 2.81; N, 9.77; Found C, 50.28; H, 2.79; N, 9.74.



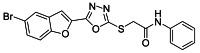



########### 3.2.8. 2-((5-(5-Bromobenzofuran-2-yl)-1,3,4-oxadiazol-2-yl)thio)-N-(2,4-dimethylphenyl)acetamide (**5h**)

Yield (69%), White Powder, MP (238–240 °C), ^1^H-NMR, DMSO-*d_6_*, 400 MHz (δ/ppm): 2.16 (s, 3H, Me), 2.24 (s, 3H, Me), 4.39 (s, 2H, CH_2_), 6.97 (d, *J* = 8.0 Hz, 1H, Ar), 7.02 (s, 1H, Ar), 7.76 (s, 1H, Ar), 7.78 (s, 1H, Ar), 7.25 (d, *J* = 8.0 Hz, 1H, Ar), 7.66 (dd, *J* = 2.0 and 8.0 Hz, 1H, Ar), 8.04 (d, *J* = 2.0 Hz, 1H, Ar), 10.75 (s, exch., 1H, NH).^13^C-NMR, DMSO-*d_6_*, 100 MHz (δ/ppm): 17.7, 20.6, 36.6, 109.8, 114.1, 116.6, 125.0, 125.2, 126.6, 130.0, 131.0, 131.8, 133.3, 134.8, 141.0, 153.9, 158.0, 164.1, 164.9.HR-MS Calcd. 458.23[M+] (100%). Anal.Calcd.for: C_20_H_16_BrN_3_O_3_S; C, 52.41; H, 3.52; N, 9.17; Found C, 52.43; H, 3.50; N, 9.20.



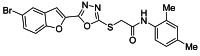



############ 3.2.9. 2-((5-(5-Bromobenzofuran-2-yl)-1,3,4-oxadiazol-2-yl)thio)-N-(2,5-dimethoxyphenyl)acetamide (**5i**)

Yield (76%), Grey Powder, MP (177–179 °C), ^1^H NMR (DMSO-*d_6_*; 400 MHz): 3.67 (s, 3H, OMe), 3.79 (s, 3H, OMe), 4.46 (s, 2H, CH_2_), 6.65 (dd, *J* = 8.0 and 2.0 Hz, 1H, Ar), 6.97 (d, *J* = 8.0 Hz, 1H, Ar), 7.63−7.76 (m, 4H, Ar), 8.02 (d, *J* = 2.0 Hz, 1H, Ar), 9.71 (s, exch., 1H, NH),^13^C NMR (DMSO-*d_6_*; 100 MHz): 37.0, 55.4, 56.3, 107.9, 108.5, 109.8, 112.1, 114.1, 116.6, 125.1, 127.8, 129.2, 130.0, 141.0, 143.5, 153.0, 153.9, 158.0, 164.1, 165.2.HR-MS Calcd. 489.33[M+] (100%). Anal.Calcd.for: C_20_H_16_BrN_3_O_5_S; C, 48.99; H, 3.29; N, 8.57; Found C, 49.01; H, 3.32; N, 8.61.



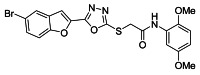



############# 3.2.10. 2-((5-Benzofuran-2-yl)-1,3,4-oxadiazol-2-yl)thio)-N,Ndiethylacetamide (**5j**)

Yield (50%), light grey powder, MP (96–98 °C) [26].



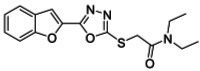



### 3.3. Tyrosinase Inhibition Assay

The bacterial tyrosinase enzyme isolation and preparation were fully described in our previously published paper and in other reported literature [58,59]. A novel synthesized series of oxadiazole-based furan molecules were screened for anti-tyrosinase activity. Tyrosinase inhibition was investigated by modifying the reported methods of Kim [60]. Briefly, phosphate buffer (0.05 M), L-tyrosine (765 µL; 2 mM) and tested compounds (35 µL) were incubated at ambient temperature for 10 min then bacterial tyrosinase (200 µL; about 48 U/mL) was added and the whole assay solution was allowed to stand at 25 °C for 2 min. The formation of dopachrome was monitored after incubation (2 min) by measuring the increase in optical density (OD) at λ_max_ 475 nm. The stock solution for the entire synthesized series of oxadiazole-based furan moieties (1 mM) was prepared in DMSO (dimethyl sulfoxide). Five different dilutions of each oxadiazole-based furan moiety was prepared. Ascorbic acid (1 mM) was utilized for the standard tyrosinase inhibitor, and the values were expressed as IC_50_, the concentration of tested samples which caused the 50% inhibition. The percentage (%) of bacterial tyrosinase inhibitory activity was determine by utilizing the formula:(2)$$\%\;Inhibition=\left[\left(A-B\right)A\right]\times 100$$

Here, *A* represents control enzyme, i.e., without inhibitor, while *B* represented the test sample, i.e., inhibitor.

### 3.4. Computational Methodology

#### 3.4.1. Retrieval of Bacterial Tyrosinase Structure from Protein Data Bank (PDB)

The PDB entry with PDB ID code 3NM8 (https://www.rcsb.org/structure/3nm8) was used to access the bacterial tyrosinase crystal structure for energy minimization with UCSF Chimera 1.10.1. The hydrophobicity graph, basic stereo-chemical properties and Ramachandran graph of bacterial tyrosinase were retrieved using the software “Discovery Studio 4.1 Client” (https://discover.3ds.com/discovery-studio-visualizer-download) and MolProbity server, respectively. The architecture and quantitative evaluation of protein structure quality of bacterial tyrosinase was performed using the online VADAR 1.8 (http://vadar.wishartlab.com/) tool [61,62].

#### 3.4.2. Designing of Ligands **5a–j** and Chemoinformatic Analysis

The ACD/ChemSketch was used to draw chemically designed furan-oxadiazole appended ligands **5a–j**, retrieved in mol format and utilized in docking process in pdb format. The online Molsoft (http://www.molsoft.com/) tool was used to predict basic biochemical properties of newly synthesized furan-oxadiazoleligands. The Lipinski’s rule of five (RO5) was applied for validation of all the synthesized ligands to check and confirm their lead-like behavior [63].

#### 3.4.3. Molecular Docking of Furan-Oxadiazoles **5a–j**

The furan-oxadiazole designed structural hybrids **5a–j** in pdb format underwent energy minimization using UCSF Chimera 1.10.1 before the molecular docking simulation procedure. Moreover, Dock Prep was used to add Gasteiger partial charges in the furan-oxadiazole ligand structures [61]. The virtual screening tool PyRx was employed for the molecular docking simulation experiments of all the furan-oxadiazole analogues against bacterial tyrosinase [64]. For the determination of good conformational position in the active region of target bacterial tyrosinase protein, the following grid box center values of X = −9.3752, Y = 5.2651 and Z = −5.4966 were used, while size values were adjusted as X = 25.0000, Y = 23.8378 and Z = 21.5235. The docking process against bacterial tyrosinase was individually performed for all the furan-oxadiazole ligands **5a–j** with a default exhaustiveness value 8. The structure-activity relationship (SAR) and lowest binding energy (kcal/mol) parameter values were applied for evaluation of predicted docked complexes. For three dimensional (3D) graphical illustrations, the Discovery Studio (V 2.1, 2008) and UCSF Chimera 1.10.1 tools were applied for all the docked complexes [65].

#### 3.4.4. ADMET and Drug-Likeness Studies

Studies involving the absorption, distribution, metabolism and excretion studies, along with drug-likeness investigations of all the compounds **5a–j,** were performed by employing SwissADME [66]. Toxicity studies of these compounds which involved carcinogenicity predictions, the Ames toxicity tes, and inhibition of the hERG potassium channel were performed using the admetSAR online server [67,68].

### 3.5. Statistical Data

The Prism software was used to analyze statistical data, while the results were measured in triplicates and depicted as mean ± SD.

## 4. Conclusions

In the present work, novel furan-oxadiazole S-alkylated amide linked hybrids **5a****–j** were designed and screened for determination of their therapeutic potential against pharmacologically important bacterial tyrosinase enzyme. Many of the synthesized compounds revealed significant activity against tyrosinase. The compound **5a** (IC_50_ 11 ± 0.25 μM) was found to be most potent and displayed the best tyrosinase inhibitory efficacy in comparison with the reference drug ascorbic acid with an IC_50_ value of 11.5 ± 0.1 μM. The compounds **5b****–5d** showed significantly good inhibition efficacy against tyrosinase enzyme (IC_50_ 12.4 ± 0.0–15.5 ± 0.0 μM). The moderate tyrosinase inhibitory activity was displayed by **5e–5h** (IC_50_ 25 ± 0.75–36 ± 0.25 μM). It was noted that among the synthesized derivatives, the least effective structural hybrids were **5i** and **5j** which exhibited IC_50_ values of 48 ± 0.96 and 49.5 ± 0.92, respectively. Based on these results, it is established that the compounds **5a–5d** containing EWD (electronegative halogen groups) showed excellent to good inhibitory efficacy as compared to furan-oxadiazole compounds **5e–5j** containing ED (ethyl, methyl, methoxy) groups. SAR studies revealed that overall substituent-related tyrosinase activity decreases in the following way: 2-flouro > 2-chloro > 3,4-dichloro > 4-fluoro > 3,4-dimethyl > 2-methoxy > phenyl > 2,4-dimethyl > 2,5-dimethoxy > diethyl. Molecular docking, cheminformatics studies, ADMET and drug-likeness studies demonstrated excellent association with the experimental findings of tyrosinase enzyme inhibition studies of novel oxadiazole-furan structural hybrids. Thus, it is anticipated that the furan-oxadiazole derivative **5a** is a more effective reagent than standard ascorbic acid. Therefore, it could be considered as potential lead molecule for the design and development of selective tyrosinase inhibitors applicable to skin whitening and to malignant melanoma anticancer agents, and with lesser side effects.

## Data Availability

Data are present within the article.

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
