# Peer review of "Ultrasonic-Assisted Synthesis of Benzofuran Appended Oxadiazole Molecules as Tyrosinase Inhibitors: Mechanistic Approach through Enzyme Inhibition, Molecular Docking, Chemoinformatics, ADMET and Drug-Likeness Studies"

_ijms, 2022, doi:10.3390/ijms231810979_

Round 1

Reviewer 1 Report

Please check the notes in the PDF file. 

The manuscript has summarized the authors' work on synthesizing a series of tyrosinase inhibitors with benzofuran and oxadiazole. Utilizing computational methods, the authors also showed the potential of these molecules in turning into active drugs.  

However, the authors should improve the consistency of the content. The introduction focused more on homo sapiens tyrosinase, while the experiment focused more on tyrosinase from Bacillus subtilis. It will be more relevant to discuss what benefit this research has in agriculture, medicine, etc. 

I think some modifications to the layout mentioned in the attached file will make the content clear. 

Overall, I believe the manuscript fits our journal, which has decent quality and rational research design. I will agree to have it published in our journal after some revision. 

Author Response

RESPONSE TO REVIEWER-1 COMMENTS

Comments No

Reviewer-1 Comments

Response

1

It is unnecessary to list all compounds synthesized in the abstract part.

We have removed the list of synthesized compounds from the abstract.  Moreover, the compounds with significant tyrosinase inhibition results have been mentioned in abstract.

2

However, the authors should improve the consistency of the content. The introduction focused more on homo sapiens tyrosinase, while the experiment focused more on tyrosinase from Bacillus subtilis. It will be more relevant to discuss what benefit this research has in agriculture, medicine, etc.

We are thankful for highlighting this discrepancy. We have corrected it and applications of bacterial tyrosinase have now been discussed in introduction with newly added references 12-17.

3

Please correct the size of the shape and font. All look stretched. Figure-1

We have corrected Figure-1

4

Please correct the size of the shape and font. All look stretched.

We have corrected all figures which were stretched.

5

Why did you choose ultrasonic assisted condition? Why does the normal condition not work? Or what advantage does the ultrasonic-assisted reaction have over normal conditions?

Conventional methodologies have generally lower yields and usually require longer reaction times. In order to address this problem, ultrasound assisted synthesis has been designed as a great alternative to conventional synthesis, since the same reactions can be carried out in lesser reaction times with higher yields. Therefore, we were interested to check the yields of the products in ultrasound assisted synthesis and as expected ultrasound assisted synthesis resulted in high yields with lower reaction times. For reference, please find some literature reports:

·       Mohamed F. Mady, Ahmed A. El-Kateb, Ibrahim F. Zeid, Kåre B. Jørgensen, "Comparative Studies on Conventional and Ultrasound-Assisted Synthesis of Novel Homoallylic Alcohol Derivatives Linked to Sulfonyl Dibenzene Moiety in Aqueous Media", Journal of Chemistry, vol. 2013, Article ID 364036, 9 pages, 2013. https://doi.org/10.1155/2013/364036

·       Rezki N, Al-Sodies SA, Shreaz S, Shiekh RA, Messali M, Raja V, Aouad MR. Green Ultrasound versus Conventional Synthesis and Characterization of Specific Task Pyridinium Ionic Liquid Hydrazones Tethering Fluorinated Counter Anions: Novel Inhibitors of Fungal Ergosterol Biosynthesis. Molecules. 2017 Nov 7;22(11):1532. doi: 10.3390/molecules22111532. 

·       Wangxin Liu, Xianliang Luo, Yang Tao, Ying Huang, Minjie Zhao, Jiahui Yu, Fengqin Feng, Wei Wei. Ultrasound enhanced butyric acid-lauric acid designer lipid synthesis: Based on artificial neural network and changes in enzymatic structure, Ultrasonics Sonochemistry, 88, 2022, 106100. https://doi.org/10.1016/j.ultsonch.2022.106100.

·       Sumaiya Tabassum, Santhosh Govindaraju, Riyaz-ur-Rahaman Khan, Mohamed Afzal Pasha. Ultrasound mediated, iodine catalyzed green synthesis of novel 2-amino-3-cyano-4H-pyran derivatives, Ultrasonics Sonochemistry, 24, 2015, Pages 1-7.

https://doi.org/10.1016/j.ultsonch.2014.12.006.

6

Remove Figure 4, as I do not see there is a need for having it.

We have removed Figure 4.

7

There is no need to have both Table 2 and Figure 5.

We have removed Figure 5 as suggested.

8

Please only highlight the two most potent inhibitors to clarify the point. Show other structures in your supporting information. (For figure 6, 7, and 8)

We have updated Figures as suggested by the reviewer and moved some figures to supplementary materials.

9

Tyrosinase’s importance in the human body has been first discussed in the introduction part. However, here, the docking experiment was done on a bacterial tyrosinase. Why do you not use a homo sapiens tyrosinase that has relationships with the diseases mentioned above? If you want to only focus on bacterial tyrosinase, you should avoid or only say a little about how tyrosinase works in the human body.

We are thankful for highlighting this discrepancy. We have corrected it and applications of bacterial tyrosinase have now been discussed in introduction with newly added references 12-17.

Reviewer 2 Report

Manuscript Title: Ultrasonic-Assisted Synthesis of Benzofuran Appended Oxadiazole Molecules as Tyrosinase Inhibitors: Mechanistic Approach through Enzyme Inhibition, Molecular Docking, Chemoinformatics, ADMET and Drug-Likeness Studies

Reviewer Recommendation: Major Revision

The main drawback of the manuscript is lacking discussion. The author has to concentrate much on discussing their obtained results. Along with this there are a few more comments listed below. 

Major comments 

1. Authors have mentioned a set of furan derivatives and drugs with oxadiazole core. Why have authors not used any of these compounds as standard one for the comparative study?

2. Did authors have any structural similarity of derivatives with standard molecules? Because the best active compound 5a and standard one shows more or less similar inhibitory effect against tyrosinase inhibition  assay. 

3. Authors requested to provide methodology for extraction of tyrosinase form isolated from Bacillus subtilis NCTC 10400 more in detail. 

4. Provide the detailed information on the range of concentration of compounds and standard ones that are used for the inhibition assay.

5. For in silico analysis authors have used protein sequence of tyrosinase from Bacillus megaterium; but in inhibitory assay evaluation it was isolated from Bacillus subtilis NCTC 10400. Why have authors not used the same strains for both experiments?

6. Did authors have checked the isolated tyrosine enzyme? If so, provide the detailed methods that are used for confirmation and validation. 

7. The docking score values are very less in case of all the derivatives than standard one. How do authors conclude that these compounds have less fluctuation and have good docking energy?

8. In wet lab study the compound 5j shows very less inhibitory activity but in molecular docking analysis it shows very good binding affinity with tyrosinase enzyme than standard one, discuss this one more in detail.

9. It is not clear the use of a program to find the location of an active site region (SiteMap) for a structure that presents a binder at the active site.

10. Did authors have noticed all the derivatives and standard compounds share the similar binding pocket of tyrosine enzyme?

11. Authors requested to keep all the derivatives in the binding pocket of the tyrosine compounds along with standard one, it could be easy for understand. 

12. Authors requested to provide detailed interpretation on molecular interaction analysis and mode of action of each derivative. 

13. Did authors check the stability of the complex? Authors requested to perform molecular dynamic simulation for complexes atleast for 100ns.

14. Figure quality is very poor.

Author Response

Major comments 

Comments No

Comments Detail

Response to Comments

1

Authors have mentioned a set of furan derivatives and drugs with oxadiazole core. Why have authors not used any of these compounds as standard one for the comparative study?

Usually used standard tyrosinase inhibitors drugs are Kojic acid, Arbutin, Azelaic acid, Ellagic acid, Tranexamic acid, Thioguanine and Ascorbic acid. All these drugs don’t have both furan and oxadiazole like structures in their cores.  Ascorbic acid has furan like core in its structure which is shown below.

2

Did authors have any structural similarity of derivatives with standard molecules? Because the best active compound 5a and standard one shows more or less similar inhibitory effect against tyrosinase inhibition  assay. 

Among standard compounds, ascorbic acid has furan like core with carbonyl moiety as in our synthesized drugs. Oxadiazole and furan derivatives-based standards do not have any structure similarity. Some of the related literature provided for reference is listed below;

Oxadiazole related papers:

1-    Lam, W. K.; Syahida, A.; Ul-Haq, Z.; Abdul Rahman, B. M.; Lajis, H. N. Synthesis and biological activity of oxadiazole and triazolothiadiazole derivatives as tyrosinase inhibitors. Bioorganic & Medicinal Chemistry Letters, 2010, 20 (12), 3755–3759doi:10.1016/j.bmcl.2010.04.067

2-    Mohd, M. F. F.; Aluwi, M.; Rullah,K.; Huhan. H.; Tan, Chan,K. M.; Tan, J. S.; Leong, S. W.; Mansor ,A. H., Yamin, B. M. and Wai, L. K. Synthesis and effects of oxadiazole derivatives on tyrosinase activity and human SK-MEL-28 malignant melanoma cells. RSC Advances, , 2016, DOI: 10.1039/C6RA12754A

3-    Sawan, L. R.;  Lanke, D. P.; Wadekar, J. B. Tyrosinase inhibitory activity, 3d qsar, and molecular docking study of 2,5-disubstituted-1,3,4-oxadiazoles. Journal of Chemistry, 2013, 2013, Article:ID 849782 http://dx.doi.org/10.1155/2013/849782

Furan related papers

1-    Jung, H.J.; Noh, S.G.; Ryu, I.Y.; Park, C.; Lee, J.Y.; Chun, P.; Moon, H.R.; Chung, H.Y. (E)-1-(Furan-2-yl)-(substituted phenyl)prop-2-en-1-one Derivatives as Tyrosinase Inhibitors and Melanogenesis Inhibition: An In Vitro and In Silico Study. Molecules 202025, 5460. https://doi.org/10.3390/molecules25225460

2-    Hu, X.; Wang, M.; Yan, Gui-R.; Yu, Mei-.; Wang, He-Y.; Hou, Ai-J. 2-Arylbenzofuran and tyrosinase inhibitory constituents of <i>Morus notabilis</i>. Journal of Asian Natural Products Research, 2012,  14,1103–1108.

DOI: 10.1080/10286020.2012.724400

3-    Xia, L.; Idhayadhulla, A.; Lee, Y. R.; Wee, Young-J.; Kim, S. H.. Anti-tyrosinase, antioxidant, and antibacterial activities of novel 5-hydroxy-4-acetyl-2,3-dihydronaphtho[1,2-b]furans. European Journal of Medicinal Chemistry, 2014, 86, 605-612 .http://dx.doi.org/10.1016/j.ejmech.2014.09.025

4-    Dige, N. C.; Mahajan, P. G.; Raza, H.; Hassan, M.; Vanjare, B. D.; Hong, H.; Hwan L. K.; Latip, J.; Seo, Sung-Y. Ultrasound mediated efficient synthesis of new 4-oxoquinazolin-3(4H)-yl)furan-2-carboxamides as potent tyrosinase inhibitors: Mechanistic approach through chemoinformatics and molecular docking studies. Bioorganic Chemistry, 2019,  92, 103201..https://doi.org/10.1016/j.bioorg.2019.103201

3

Authors requested to provide methodology for extraction of tyrosinase form isolated from Bacillus subtilis NCTC 10400 more in detail. 

Tyrosinase enzyme preparation, extraction and purification steps were performed using the following literature reports cited in our manuscript [58-59] as well;

1-    Elsayed, E. A.; Danial. E. N. Isolation, Identification and Medium Optimization for Tyrosinase Production by a Newly Isolated Bacillus subtilis NA2 Strain. Journal of Applied Pharmaceutical Science, 2018, 8, 093-101. https://japsonline.com/admin/php/uploads/2725_pdf.pdf

2-    Hussain, F.; Kamal, S.; Rehman, S.; Azeem, M.; Bibi, I.; Ahmed, T.; Iqbal, Hafiz M. N. Alkaline Protease Production Using Response Surface Methodology, Characterization and Industrial Exploitation of Alkaline Protease of Bacillus subtili ssp, Catal Lett, 2017, 147, 1204–1213. doi:10.1007/s10562-017-2017-5

4

Provide the detailed information on the range of concentration of compounds and standard ones that are used for the inhibition assay.

We have added this detailed information in the section 3.3 of the manuscript.

5

For in silico analysis authors have used protein sequence of tyrosinase from Bacillus megaterium; but in inhibitory assay evaluation it was isolated from Bacillus subtilis NCTC 10400. Why have authors not used the same strains for both experiments?

Thank you very much for raising this comment, well our major focus was on inhibitory potential. We could not find the required structure from PDB, and prior work was also based on using this protein structure for docking experiment for tyrosinase (Int. J. Mol. Sci. 2018, 19, 690; Biomed Res Int. 2022 Jan 11;2022:1040693.).

6

Did authors have checked the isolated tyrosine enzyme? If so, provide the detailed methods that are used for confirmation and validation. 

The detailed methods used for confirmation and validation were based on our previously published article;

1-    Hussain, F.; Kamal, S.; Rehman, S.; Azeem, M.; Bibi, I.; Ahmed, T.; Iqbal, Hafiz M. N. Alkaline Protease Production Using Response Surface Methodology, Characterization and Industrial Exploitation of Alkaline Protease of Bacillus subtilis sp, Catal Lett, 2017, 147, 1204–1213. doi:10.1007/s10562-017-2017-5

7

The docking score values are very less in case of all the derivatives than standard one. How do authors conclude that these compounds have less fluctuation and have good docking energy?

Yes, we agree with the reviewer. Our system showed less energy in comparison to standard. The fluctuations were just based on binding behavior through hydrogen or hydrophobic interactions.

8

In wet lab study the compound 5j shows very less inhibitory activity but in molecular docking analysis it shows very good binding affinity with tyrosinase enzyme than standard one, discuss this one more in detail.

The IC50 values are more variable than the docking energy values. Docking study was just employed to check the interactive profile against target protein.

9

It is not clear the use of a program to find the location of an active site region (SiteMap) for a structure that presents a binder at the active site.

Basically, active region was identified and confirmed through already published literature, as we have mentioned in the manuscript.

10

Did authors have noticed all the derivatives and standard compounds share the similar binding pocket of tyrosine enzyme?

We had just focused on our most potent compounds that showed high inhibitory potential against target protein.

11

Authors requested to keep all the derivatives in the binding pocket of the tyrosine compounds along with standard one, it could be easy for understand. 

We are agreeing with reviewer comment, however, our major focus was on most potent compound. Therefore, we considered this ligand for interaction analysis.

12

Authors requested to provide detailed interpretation on molecular interaction analysis and mode of action of each derivative. 

As we had discussed in prior comment, our major emphasis was on compound 5a; therefore, we considered this compound in our in silico analysis.

13

Did authors check the stability of the complex? Authors requested to perform molecular dynamic simulation for complexes at least for 100ns.

Thank you very much for this comment, but currently we cannot perform MD simulation experiment due to the computer hardware upgrade.

14

Figure quality is very poor.

Docking figure has been revised with 300 dpi resolution.

Round 2

Reviewer 2 Report

Author refined manuscript is in good quality to get published, I recommend it

Best wishes for the authors